# Demonstrating the Applicability of Smartwatches in PM_2.5_ Health Impact Assessment

**DOI:** 10.3390/s21134585

**Published:** 2021-07-04

**Authors:** Ming-Chien Mark Tsou, Shih-Chun Candice Lung, Chih-Hui Cheng

**Affiliations:** 1Research Center for Environmental Changes, Academia Sinica, Taipei 115, Taiwan; marktsou09@gate.sinica.edu.tw (M.-C.M.T.); chihhui104@gate.sinica.edu.tw (C.-H.C.); 2Department of Atmospheric Sciences, National Taiwan University, Taipei 106, Taiwan; 3Institute of Environmental and Occupational Health Sciences, National Taiwan University, Taipei 100, Taiwan

**Keywords:** wearable smart devices, smartwatches, wearables, personal heart rate monitoring, photoplethysmography, portable PM sensing devices, particles and health

## Abstract

Smartwatches are being increasingly used in research to monitor heart rate (HR). However, it is debatable whether the data from smartwatches are of high enough quality to be applied in assessing the health impacts of air pollutants. The objective of this study was to assess whether smartwatches are useful complements to certified medical devices for assessing PM_2.5_ health impacts. Smartwatches and medical devices were used to measure HR for 7 and 2 days consecutively, respectively, for 49 subjects in 2020 in Taiwan. Their associations with PM_2.5_ from low-cost sensing devices were assessed. Good correlations in HR were found between smartwatches and certified medical devices (*r_s_* > 0.6, except for exercise, commuting, and worshipping). The health damage coefficients obtained from smartwatches (0.282% increase per 10 μg/m^3^ increase in PM_2.5_) showed the same direction, with a difference of only 8.74% in magnitude compared to those obtained from certified medical devices. Additionally, with large sample sizes, the health impacts during high-intensity activities were assessed. Our work demonstrates that smartwatches are useful complements to certified medical devices in PM_2.5_ health assessment, which can be replicated in developing countries.

## 1. Introduction

Smartwatches have become popular over the past decade due to their real-time health monitoring functionality, including that for heart rate (HR). Around 68.6 million smartwatches were sold in 2020, and this figure is expected to reach 157.2 million by 2026 [1]. As the general population is increasingly recognizing smartwatches from advertising, consumer-grade smartwatches have been used in health research for HR monitor since 2014 [2,3]. Many types of smartwatches have an optical HR sensor that uses photoplethysmography (PPG) to measure how much blood the heart is pumping under the surface of the skin, since PPG is a low-cost, simple, and portable technology [4].

Some studies have examined the accuracy of wearable PPG sensors compared to a reference method, i.e., electrocardiogram (ECG), during different types of activities [3,5,6,7]. Although the concordance of HR measurements usually decreases as physical activity intensity increases for most brands of smartwatches [8], it is difficult to assess how smartwatches function during different activities. Despite these problems, smartwatches may be useful for longer HR monitoring periods given their low interference with daily activities [6,9].

Particulate matter with an aerodynamic diameter of less than or equal to 2.5 μm (PM_2.5_) may affect the balance of the automatic nervous system, which may then cause an increase in HR and a decrease in heart rate variability (HRV) [10]. A meta-analysis including studies between 1990 and 2017 indicated that HR is a predictor of cardiovascular morbidity and mortality in the global population or in patients with cardiovascular and cerebrovascular diseases, even after adjustment for the most important cardiovascular risk factors [11]. In addition, according to a recent review paper, an increase in the resting HR is associated with increased risk of incident hypertension, especially in individuals with an HR > 80 bpm [12]. The Gutenberg Health Study (GHS), a population-based prospective cohort study in Midwestern Germany, also indicated that increased risk is present in subjects with a HR > 64 bmp, with a hazard ratio of 1.29 per 10 bpm increase in HR [13]. Chen et al. [14] also indicated that every beat increase in HR at baseline was associated with a 3% higher risk for all-cause death, 1% higher risk for CVD, and 2% higher risk for CHD in the general male population. Therefore, in addition to the HRV, the HR seems to be an indicator of adverse impacts of PM_2.5_.

Some panel studies demonstrated that enhanced HR is associated with increased PM_2.5_ as assessed by certified medical devices [15,16]. However, subjects usually wore these devices only for 1 to 2 day, due to the skin irritation caused by the electrode gel. It would be convenient if we would be able to apply smartwatches in PM_2.5_ health studies. A previous study showed that HR data obtained by smartwatches were not numerically equal to those obtained by certified medical devices [6]. However, as long as the estimated health damage coefficients, which are defined as the coefficients of adverse health impacts caused by PM_2.5_, indicate the same direction and show similar magnitudes in the regression models as those obtained via certified medical devices, smartwatches would still be useful complements for those vulnerable or high-exposure subpopulations who may be unable to wear traditional HR monitors.

Compared to certified medical devices, wearable smartwatches with a PPG sensor are more comfortable over a longer monitoring period. HR data collection is benefitted if it can be achieved under conditions with minimal interference from the monitoring technology. Until now, no studies have used smartwatches with a PPG sensor to evaluate the PM_2.5_–HR relationship. Therefore, we aimed to assess the applicability of smartwatches in PM_2.5_ health assessment by evaluating whether smartwatches are good complements to certified medical devices for PM_2.5_ health studies, especially for developing countries. If smartwatches are affordable complements to certified medical devices, they will facilitate environmental health research in developing countries. Specifically, we wanted to assess (1) whether the tendency of the estimated PM_2.5_–HR health damage coefficients (direction of impacts) are the same when assessed via a certified ECG device versus a commercial smartwatch, and (2) whether the magnitudes of the estimated damage coefficients are similar when using certified medical devices versus smartwatches for healthy adults. In addition, the relationships between PM_2.5_ and HR were evaluated across different activity levels to demonstrate the advantages of large sample sizes to enable the assessment of the health impacts in more subpopulations in different categories of exposures or demographics.

## 2. Materials and Methods

### 2.1. Study Population

A total of 49 subjects living in an urban community within a 1 km radius in Southern Taiwan were recruited in 2020. The inclusion criteria were (1) being 40 to 75 years old, (2) being a non-smoker, and (3) having no history of cardiovascular disease (CVD). Among these 49 subjects, 55.1% were aged 40–64 years, 59.2% were women, and 61.2% had a BMI ≥ 24 kg/m^2^. Of the total number of subjects, 34.7% were housewives or retired, 16.3% were factory workers, 18.4% were office workers, and the rest worked in the service industry. The subjects were asked to follow their daily routines. Written informed consent and answers to a questionnaire regarding their socio-demographic characteristics and living environment were provided by each subject prior to monitoring. This study was reviewed and approved by the Institutional Review Board of Academia Sinica (IRB No.: AS-IRB-BM-18053).

### 2.2. PM Monitoring

For 7 consecutive days, each subject was asked to carry a small (135 × 70 × 40 mm) and lightweight (153 g) low-cost sensing (LCS) device, i.e., AS-LUNG-P (Academia Sinica-Lung-Portable version, integrated by our team), for personal PM_2.5_ and temperature monitoring. A light-scattering-based PM_2.5_ sensor (PMS3003, Plantower, Beijing, China), a temperature/humidity senor (SHT31, Sensirion AG, Staefa ZH, Switzerland), and a 3-axis acceleration sensor (ADXL345B, Analog Devices, Inc., Norwood, MA, USA) were integrated into the AS-LUNG-P. To increase mobility, AS-LUNG-P was connected with a mobile battery as a power supply. In addition, AS-LUNG-P and the mobile battery were carried in a customized bag by the subjects. Detailed specifications and the performances of AS-LUNG-P were published previously [16,17,18].

Each AS-LUNG-P was evaluated in side-by-side comparisons with the research-grade instrument, GRIMM Model 1.109 (GRIMM Aerosol Technik Ainring GmbH & Co, Ainring, German), in a hood or chamber in the laboratory before application in the field according to the procedures described earlier [18,19]. The correction equations for AS-LUNG-P were obtained under the conditions of 27.0–31.6 °C and 60.9–74.4% relative humidity (RH), obtaining an R^2^ of 0.99 [18]. In addition, the converted measurements of AS-LUNG sets were compared with data from the side-by-side GRIMM instruments in the field under the conditions of 25.9–40.9 °C and 43.4–93.8% RH, obtaining an R^2^ of 0.86 [19]. These results indicated the correction equations obtained in the laboratory were applicable in the field. The ranges of temperature and humidity in the laboratory and in previous field evaluation are close to the ranges in this study (14.5–42.6 °C and 28.3–98.7% RH). Thus, it was appropriate to convert our AS-LUNG measurements to research-grade data with correction equations obtained in the laboratory conditions. As the R^2^ values of the correction equations (ranging from 0.1 to 200 μg/m^3^) were high (0.983–0.993), the feasibility of using the AS-LUNG-P with data correction was indicated. All the AS-LUNG-P data used in this work were converted to research-grade data using their respective correction equations.

### 2.3. HR Monitoring

Each subject was asked to wear a smartwatch (Garmin Forerunner 35, Garmin Ltd., Olathe, KS, USA) on their left wrist for HR monitoring for 7 consecutive days. The Garmin Forerunner 35 is a commercially available smartwatch [20] that employs optical HR measurement technology, i.e., PPG with green LED lights on the bottom. The sampling rate increases when an activity is detected, and varies depending on the level of activity [21]. After uploading the recorded HR data to Garmin Connect via the internet, the HR data can be downloaded from the application program’s interface.

During the same monitoring period, the subjects were asked to wear ECG monitor devices (RootiRx, Rooti Labs Ltd.,Taipei, Taiwan) for 2 consecutive days. In other words, there were two days that subjects wore both smartwatches and RootiRx. RootiRx is a novel ECG patch monitor device with certifications from the EU, the US, and Taiwan [22]. A previous study demonstrated high correlation (98%) in measured beat per minute between RootiRx and a standard 12-lead Holter in 33 healthy subjects [23]. When a monitoring session finished, the recorded ECG data were uploaded via WiFi. After that, data were automatically analyzed using the empirical proprietary algorithms of Rooti Labs Ltd. to calculate the HR data. Research scientists in Rooti Labs Ltd. reviewed each batch of data and discussed abnormal observations with our team immediately. In our previous work, two subjects were removed from the datasets since we found that this subject had certain heart conditions. In the current work, no subject was removed due to abnormal conditions. The sampling rate of the RootiRx was 500 Hz.

A time-activity diary (TAD) was completed in 30 min intervals by each subject during the monitoring period concerning microenvironments, ventilations, activities, and PM-related exposure sources. The activities recorded were essential to this study, and were categorized as follows: resting (without doing any activity), commuting, working, cooking, worshipping, shopping, exercising (jogging, cycling, workouts, etc.), eating, bathing/showering, sedentary activities (watching television or movies, reading books, playing sedentary video games, chatting, etc.), and other activities (the participants were asked to specify). Every subject’s TAD was checked daily by a well-trained interviewer.

### 2.4. Statistical Analysis

Data analysis was performed using R Version 3.5.2 (The R Foundation, Austria, 2018). The 5 min means of measurements were used in this study. For each 5 min segment, the mean HR measurements <40 and >200 beats/min were excluded to reduce the bias caused by artifacts in HR measurements. The final sample sizes of the 5 min mean measurements were 63,061 and 15,294 over 7 and 2 days, respectively, after excluding data during sleeping and raining hours.

Previous works demonstrated that activity is an important factor in PM_2.5_ health assessments [16,17]. Therefore, the following models incorporate activity intensity, calculated using Equation (E.1):(E.1)$$\mathrm{Activity}\;\mathrm{intensity}=\sqrt{X^{2}+Y^{2}+Z^{2}}$$where *X, Y*, and *Z* are the maximum accelerations in the left–right, cranio–caudal, and dorso–ventral dimensions, respectively, obtained from the motion sensors in the RootiRx and AS-LUNG-P over 7 and 2 days, respectively. As the Garmin Forerunner 35 does not have a built-in motion sensor, the activity intensity measurements obtained from the AS-LUNG-P were used in the model for assessing the impacts of PM_2.5_ on HR obtained from Garmin Forerunner 35 as an adjustment term. Notably the AS-LUNG-P was carried in a bag located near the chest, instead of being attached to the body like the RootiRx. Therefore, the movements of the AS-LUNG-P may be different to those of the RootiRx, which likely measured the real activity levels of the subjects. Therefore, a correction equation was established to convert the activity intensity measurements obtained from AS-LUNG-P to RootiRx-comparable measurements to examine their correlation. Additionally, the subjects were allowed to place the AS-LUNG-P on a nearby table when they were engaged in activities in the same environment, such as reading, watching television, sleeping, etc. The underestimation of activity intensity with AS-LUNG-P on nearby tables needed to be removed; the activity intensity measurements during the sleeping period were used as a proxy to remove these underestimates. We found that the activity intensity measurements during sleeping periods were mostly below 1100 mG (99.4%) when the AS-LUNG-P was placed on a table (Appendix A). To avoid these errors, activity intensity measurements below 1100 mG were excluded from the subsequent correction equations for the RootiRx and AS-LUNG-P. The correction equation is shown in the Supplementary Information (Appendix A).

According to the results of our previous work [16,17], demographic factors are important in assessing the health impacts of PM_2.5_. Therefore, the Wilcoxon rank sum test was used to compare the PM, HR, and activity intensity data from different demographic groups due to the skewed distributions. In addition, the HR measurements from smartwatches may differ between activities undertaken [6]. Spearman’s rank correlation coefficients (*r_s_*) were used to examine the correlations between the Garmin Forerunner 35 HR (G-HR) and the RootiRx HR (R-HR) throughout various daily activities, as obtained from TADs.

Based on our previous work [16], a generalized additive mixed model (GAMM) was applied to assess the associations between PM_2.5_ concentration and HR data, as in Model (M.1):(M.1)$$\log\left(\mathrm{HR}\right)=\beta_{0}+\beta_{1}{\mathrm{PM}}_{2.5}+\beta_{2}\mathrm{Age}+\beta_{3}\mathrm{Gender}+\beta_{4}\mathrm{BMI}+\beta_{5}\mathrm{Temperature}+f_{1}\left(\mathrm{AI}\right)+f_{2}\left(\mathrm{Time}\right)+\gamma_{\mathrm{subject}}+\varepsilon$$where HR refers to G-HR used for 7 day or R-RH used for 2 day monitoring, which were log10-transformed to improve their normality; β_0_ is the intercept; β_i_ denotes the regression coefficients of PM_2.5_, age (40 to 64 years or 65 to 75 years), gender (male or female), body mass index (BMI; normal-weight and overweight/obese groups), and temperature; and *f*_j_ denotes the smooth functions of activity intensity (thin plate spline function) and time of day (penalized cubic regression spline function). According to the definition of the Health Promotion Administration in Taiwan, BMI < 24 kg/m^2^ was defined as the normal-weight group and BMI ≥ 24 kg/m^2^ was defined as the overweight/obese group. In addition, the first-order autocorrelation of time of day was controlled in the model. γ is the random effect of the subjects, which was controlled in the model to account for individual difference, and ε is the error term. The effects of PM_2.5_ on HR are presented as percentage changes per interquartile range (IQR) increase, and 95% confidence intervals (CI) are included. Our results were considered to be statistically significant if the *p*-value was <0.05.

To further assess the associations between the PM_2.5_ and G-HR data captured during activities of different intensities, the activities included in the TAD were categorized into three groups: (1) resting, (2) low-intensity, and (3) moderate- to high-intensity activities. For the resting group, the data obtained during sleep were excluded due to the lack of actual activity intensity data available from the AS-LUNG-P, which was placed on the bedside during sleep. The low-intensity activities included commuting, working, cooking, worshipping, shopping, eating, bathing/showering, sedentary activity, and other activities. Finally, exercising was assigned to the moderate- to high-intensity group. The activity intensity measurements were 2120 ± 1310, 2090 ± 1080, and 3030 ± 1850 mG for resting, low-intensity, and moderate- to high-intensity activities, respectively. This activity evaluation was essential for determining whether PM_2.5_–HR relationships should be assessed separately during engagement in activities of different categories. This detailed evaluation was possible since smartwatches collect larger sample sizes of HR data than the medical devices used.

## 3. Results

### 3.1. PM_2.5_ Concentration, HR, and Activity Intensities

According to the time-activity diary, the percentage of the time that subjects did not carry the AS-LUNG-P was very low (2.4%). Among these 49 subjects, there were 10 subjects with complete data during the entire monitoring period (20.4%). For those subjects who had lost partial data (*n* = 39), 37 subjects (94.9%) had a loss rate <6%, with the highest loss rate being 0.43%. The highest loss rate for AS-LUNG-P was found in a female subject (0.43%) due to the inconvenience of carrying the AS-LUNG-P when working in the kitchen. In addition, the overall data collection rates were 96.2% and 94.4% for R-HR and G-HR, respectively.

Table 1 shows the PM_2.5_ concentration, HR, and activity intensity according to the subjects’ characteristics. In the 7 day monitoring period, the overall 5 min means of PM_2.5_ concentration, G-HR, and activity intensity from the AS-LUNG-P were 21.5 ± 11.6 μg/m^3^, 85.1 ± 16.6 bpm, and 1060 ± 1150 mG, respectively. In the 2 day monitoring period, the overall R-HR and activity intensity of the RootiRx were 81.9 ± 12.3 bpm and 1930 ± 440 mG, respectively. The results indicate that, in general, the personal PM_2.5_ concentration from G-HR and R-HR (Table 1a), and the activity intensity from AS-LUNG-P and RootiRx (Table 1b), were statistically significantly different between the different age, gender, and BMI groups. Therefore, these factors were included in the subsequent GAMM analysis.

### 3.2. Correlations between G-HR and R-HR

Figure 1 shows the correlations between G-HR and R-HR during various daily activities, including all activities, resting, commuting, working, cooking, worshipping, shopping, exercising, eating, bath/shower, sedentary activities, and other activities (Figure 1a–l, respectively). Overall, the correlations between G-HR and R-HR were over 0.5 during various activities. The highest correlation coefficient arose when subjects were resting (*r_s_* = 0.816). Other activities had the second highest correlation coefficient (*r_s_* = 0.797), probably because other activities were mainly included the low-intensity activities, such as waiting to see the doctor in the clinic and housekeeping. Therefore, the correlation coefficient of the other activities was similar to that of sedentary activities (*r_s_* = 0.741). The correlation between the G-HR and R-HR was relatively low when the subjects were exercising. This may be because the smartwatches were too loose during exercise, resulting in underestimation of HR. The correlation analysis indicated that the data from G-HR were generally highly correlated to those from R-HR when the subjects were engaged in their daily routines, except when they were exercising.

### 3.3. Evaluation of Impacts of PM_2.5_ on G-HR and R-HR

Table 2 displays the results from the GAMM analysis of PM_2.5_ impacts on HR in a 5 min resolution. Comparing the models of R-HR and G-HR, the coefficients for the impacts of PM_2.5_ showed the same tendencies, i.e., positive associations. Additionally, the tendencies in the adjustment term that had statistical significance, i.e., BMI, were the same. For R-HR, an increase in personal PM_2.5_ exposure of one IQR (7.4 μg/m^3^) was associated with an increase of 0.229%, so that an increase in PM_2.5_ exposure of 10 μg/m^3^ was associated with an increase of 0.309% in R-HR. G-HR produced similar results. An increase in personal PM_2.5_ exposure of one IQR (8.3 μg/m^3^) was associated with an increase of 0.234%, so that an increase in PM_2.5_ exposure of 10 μg/m^3^ was associated with an increase of 0.282% in G-HR. In short, the difference in percentage change between G-HR and R-HR was 8.74%, (0.309–0.282%)/0.309% × 100, when the PM_2.5_ exposure increased by 10 μg/m^3^. Likewise, the BMI coefficients were not particularly different between these two models. The results of the GAMM analysis indicated that the coefficients of the effects of PM_2.5_ on HR, as assessed by a smartwatch, showed the same (positive) trend as the coefficients assessed by certified medical devices. In addition, the magnitudes of these coefficients assessed by a smartwatch were similar to those measured via certified medical devices. Compared to the certified medical devices, smartwatches cause less disturbance in the daily routine of the subjects. The subjects were willing to wear them for a longer period of time (in our case, 7 days), thus providing more data, increasing the sample size, and thus the subsequent statistical power of the data analysis. Figure 2 displays the relationships between PM_2.5_, activity intensity, and G-HR from the models adjusted for temperature, activity, and time of day. The example shown in Figure 2a is from a subject with exercising events accounting for approximately 3% of the total time; while Figure 2b is an example of a subject without exercising events. These two graphs clearly showed different patterns of the relationships of PM_2.5_, activity intensity, and HR. The overall relationship of these three variables for all subjects is shown in Figure 2c; higher G-HR measurements were generally found when subjects exposed to higher PM_2.5_ levels with a higher activity intensity. The underlying physiological mechanisms of this different patterns need to be further explored. Nevertheless, smartwatches are useful complements to certified medical devices for PM_2.5_ health evaluations.

We further evaluated the associations between personal PM_2.5_ exposure and G-HR during activities of various intensity levels (Table 3); this analysis was not conducted with R-HR due to the limited sample size. Most activities in which the subjects engaged were low-intensity activities. The results indicated that the elevated PM_2.5_ concentration was significantly associated with G-HR for low-intensity activities (*n* = 23,909), and the elevated PM_2.5_ concentration was marginally significantly associated with G-HR for moderate- to high-intensity activities (*n* = 764). In addition, a percent increase in G-HR per 10 μg/m^3^ of PM_2.5_ exposure increase for moderate- to high-intensity activities (4.53) was one order of magnitude larger than that for low-intensity activities (0.219), which is close to the result derived without activity stratification. Our results showed that the impact of different degrees of PM_2.5_ on HR can be observed depending on the intensity of the activities, except for resting.

## 4. Discussion

Smartwatches are non-invasive and easily accessible consumer-grade devices. They can continuously monitor HR during daily routines with minimal interference. This study provided evidence that smartwatches are useful complements to certified medical devices for PM_2.5_ health evaluations, showing the same (positive) trend and a similar magnitude in the damage coefficients for the effects of PM_2.5_ on HR. The coefficient difference was within an acceptable range (10%). To the best of our knowledge, this is the first study demonstrating similar associations between personal PM_2.5_ levels and HR when assessed via smartwatches compared with via certified medical devices.

Some studies have evaluated the accuracy of smartwatch HR measurements compared to those from ECG when participating in different types of activities [3,5,6,7,9,24]. The activities usually assigned by researchers have varied (such as running on the treadmill at different speeds ranging from 3.2 to 9.6 km/h [25]) or recorded using a diary for their daily activities (such as sitting, walking, running, and activities of daily living [6]). The results showed that smartwatches may under- or overestimate the HR measurements. It was also found that the accuracy of smartwatches during rest and low-intensity activities was generally higher than that during high-intensity activities for most brands of smartwatches [8,9,26]. In addition, these studies indicated that compared to certified medical devices, smartwatches may generally underestimate HR during vigorous activities. This was explained by the increased degree of erratic wrist movements during vigorous activities [6]. This is consistent with the results of the current study. The data collection rate was 98.4% during exercising from 26 subjects (based on 1 min resolution). Almost all subjects (25 of 28 subjects) collected complete data during exercising in this study, but the lowest correlation between G-HR and R-HR was observed during exercise. G-HR sometimes provided underestimation during exercise, probably because the smartwatches were loosened and could not accurately monitor HR. Moreover, the accuracy of HR measurements depends not only on the type and intensity of the physical activity, but also on the user’s physical characteristics and fit of the tracker [21]. Nevertheless, the potential problems did not affect our study. We tested the smartwatches under different activities prior to the field campaign. Notably, the smartwatches used in this study worked well and we did not observe any malfunctions during high-intensity activities. Thus, all data collected regardless the activity levels were used in the PM_2.5_–health evaluation.

Most of the previous studies assigned scheduled activities to their subjects, and so could not capture 24 h HR under real-life conditions. Only one study evaluated the accuracy of HR measurements captured by a smartwatch during a continuous 24 h period [6]. In that study, the overall accuracy of the HR data obtained from smartwatches was acceptable (<10% mean absolute percent error relative to the ECG). However, they suggested that any single measurement in real time cannot be used as an accurate measurement for medical purposes given the presence of some outliers in smartwatch measurements. This is consistent with our study. In addition, the overall correlation coefficient in this study was generally higher than for any specific activity except resting, working, sedentary activities, and other activities. This is probably because the correlation coefficients for these four activity groups were >0.72, accounting for approximately 70% of the data. The overall G-HR data were moderately to highly correlated with those from R-HR, whereas there were a number of outliers observed, especially for higher HR periods, which occurred usually during high-intensity activities. Nevertheless, the HR measurements from smartwatches showed good correlation with those from certified medical devices under most conditions, especially the low-intensity activates. Notably, most activities were low-intensity in the daily routines. For example, subjects spent less than 2% of total time exercising in this study. Thus, the HR measurements obtained from smartwatches can be used in PM_2.5_–health evaluation for the general population who engage in daily routines. Even though smartwatches may underestimate HR, the impacts of PM_2.5_ on HR can still be observed through them, and the estimated PM_2.5_ damage coefficients showed similar magnitudes, with differences of only <10%. In addition, compared to R-HR, G-HR had a wider 95% CI range of estimated PM_2.5_ damage coefficients.

Another earlier study integrated smartwatches with other low-cost sensors to concurrently monitor HR and other environmental factors; one study used an LCS device, a smartphone and a smartwatch to measure ambient CO_2_, noise levels, and HR, respectively [27]. Some outliers in the HR measurements (5%) were observed, which were then removed via visual data inspection. The authors did not compare the HR measurements obtained with the smartwatch to those obtained with certified medical devices to assess their accuracy. We recorded HR measurements during the same monitoring period with both smartwatches and certified medical devices, and compared their results concerning the health impacts of PM_2.5_. Our work demonstrated that, with a sufficient sample size, smartwatches not only displayed the same (positive) trend in the coefficients of PM_2.5_ as the certified medical devices, but also showed similar magnitudes of the impacts of PM_2.5_ to those derived from certified medical devices, with a difference of only <10%. In other words, the large sample size of HR measurements may outweigh the drawbacks of irregular measurements in assessing the PM_2.5_–HR relationship. This work demonstrated that low-cost smartwatches are useful complements to medical devices in environmental health studies, especially in developing countries where PM_2.5_ pollution is severe and resources are limited.

Both HR and HRV are the indicators of autonomic function according to Task Force of the European Society of Cardiology the North American Society of Pacing Electrophysiology [28]. Some studies recently also found that elevated PM_2.5_ exposure was associated with both decreased HRV and increased HR, focusing on daily routines or a certain period such as cycling [15,16,17,29,30,31]. Moreover, the associations between exposure to PM_2.5_ and increased HR were found in mice [32,33,34]. Therefore, HR is a useful indicator for autonomic nervous system, with increased HR indicating the adverse impacts from air pollutants such as PM_2.5_.

Moreover, some recent studies assessed the relationship of PM_2.5_ and HR focusing on resting HR. In a trial of evaluating the intervention of B vitamin supplementation to attenuate the acute autonomic effects of PM_2.5_, 2 h PM_2.5_ exposure (250 μg/m^3^) was associated with 3.8 bpm higher resting HR for the control group [35]. Another cross-sectional study for 10 million reproductive-age adults in China found that an increase in 3-year average PM_2.5_ exposure of 10 μg/m^3^ was associated with an increase of 0.076 bpm in the resting HR [36]. For other studies assessing the relationships between PM_2.5_ and HR; they typically did not differentiate the Resting HR and HR. Nevertheless, these studies found the similar trends (associations between elevated PM_2.5_ exposure and increasing HR) as those from studies focusing on resting HR.

In this study, an increase in PM_2.5_ exposure of 10 μg/m^3^ was associated with an increase of 0.282% in G-HR. Previous studies have demonstrated that short-term exposure to PM_2.5_ is associated with increased HR, ranging from 0.43 to 4.8% per 10 μg/m^3^ increase PM_2.5_ exposure [16,30,31]. Zhang et al. [31] indicated that although the impacts of PM_2.5_ on HR measurements may be small, it still a significant public health issue, especially for the susceptible population or patients with cardiovascular diseases. Cardio function may also be affected by other factors, such as other air or other environmental pollutants [15,30] and preexisting medical conditions [37]. Further studies should clarify how these factors modify the impacts of PM_2.5_ on HR measurements.

With large sample sizes, we further evaluated the impacts of PM_2.5_ on G-HR during activities of various intensities. Although G-HR may provide considerable underestimations of HR during moderate- to high-intensity activities, the results still showed the greater impacts (one order of magnitude) of PM_2.5_ on G-HR for moderate- to high-intensity activities compared to low-intensity activities. The higher PM_2.5_ impacts on subjects during higher-intensity activities may be expected but few studies have quantified the impacts on different activity levels due to the limited sample sizes in previous studies [38]. Increased physical exertion may increase the inhalation and lung deposition of PM_2.5_, which result in elevated HR [39]. Smartwatches may have higher variation in HR measurements than the certified medical devices and these smartwatches may not be state-of-the-art; however, these smartwatches can collect a large sample size to differentiate the health damage coefficients among specific activities. For example, even though smartwatches may have underestimated HR measurements during moderate- to high-intensity activities in this study, the sufficient sample size due to the extended monitoring period may reduce the impacts of uncertainty derived from the smartwatches. In addition to the difference in the activity intensity measurements, the HR measurements were also different in different levels of activity groups. The mean HR measurements were 77.2 ± 14.5, 85.8 ± 16.3, and 92.7 ± 19.1 bmp for resting, low-intensity activities, and moderate- to high-intensity activities, respectively. Therefore, the categories of the level of activity intensity were considered an additional contribution from daily activities to HR measurements. With smartwatches collecting sufficient data, the much higher impacts of PM_2.5_ on HR during high-intensity activities could be differentiated from the lower impacts during low-intensity activities. This demonstrates the great advantages of using smartwatches in environmental health studies. These results have important implications for future PM_2.5_–HR studies, which should differentiate their assessment according to different activity levels, and for the formulation of practical health-promotion tactics, which should emphasize the greater HR impacts of PM_2.5_ during exercise to alert athletes and regular runners/cyclists to not exercise in polluted areas.

The low correlation coefficient of the activity intensity measurements between AS-LUNG-P and RootiRx (*r* = 0.409) resulted in higher variability, which may bias the health damage coefficient of PM_2.5_ toward null (insignificance). In this study, the Garmin Forerunner 35 smartwatches did not have a motion sensor. Therefore, we used the measurements recorded by the motion sensor of AS-LUNG-P for activity intensity in the models. The devices used in this study were carried differently by the subjects. RootiRx was directly attached to the chest, whereas AS-LUNG-P was carried in a bag near the chest. The movements of these two devices were somewhat different in the different activities. For example, when subjects were walking, the activity intensity measured by RootiRx in the chest may represent the actual movement of the subjects, whereas that measured by AS-LUNG-P in a bag near the chest may measure the subjects’ movement plus the extra movements due to bag swing, introducing higher variability, shown as higher standard deviations in Table 1b. Therefore, the activity intensity measured by RootiRx was used in the PM_2.5_–HR relationship assessment in this study. However, the RootiRx was carried by the subjects for only 2 days due to potential skin irritation. In this study, we established a correction equation to convert the activity intensity measurements obtained from AS-LUNG-P to RootiRx-comparable measurements. Even though the high variability in the corrected RootiRx-comparable measurements may have caused the health damage coefficient of PM_2.5_ to become insignificant, our results still showed significant associations between PM_2.5_ and G-HR, with the same (positive) trend and a similar magnitude in the damage coefficients of PM_2.5_ on HR compared with those assessed via certified medical devices. Therefore, it could be concluded that, even though the activity intensity measurements involved higher uncertainty, the large HR sample size collected via smartwatches overcomes this uncertainty. The combination of a low-cost PM_2.5_ sensor (AS-LUNG-P) and a smartwatch provided similar health damage coefficient estimates in this work. This again demonstrates the feasibility of applying smartwatches in environmental health research.

Due to the aforementioned advantages, we chose to use smartwatches in our study rather than other smart devices. In addition, compared to other smart devices, smartwatch technology is more mature. For example, smart clothing also has been used to monitor HR. Buregeya et al. [30] used a smart T-shirt to monitor cardiac function and evaluate its association with traffic-related PM_2.5_ exposure. The smart T-shirt enabled the measurement of ECG parameters during cycling. Smart clothing can also integrate various physiological indicators of the human body, including ECG sensors. However, smart clothing still faces some challenges, including the interference effects of the human body on signals [40]. Conversely, the efficacy of the green-light PPG sensors used in this study was evaluated as interfering less in the tissue and vein region in different environmental temperatures [41].

The primary objective of this study was to find an affordable complement to the certified medical devices in environmental health research for developing countries; thus, we did not select the most state-of-the-art smartwatches, which are much more expensive. We selected a model of smartwatch with HR measurement, which is a typical function for health monitoring of almost all types of smartwatches. They are relatively affordable for scientists in developing countries where resources are limited. In addition, we selected a model of smartwatch without an accelerometer, since AS-LUNG is already equipped with a G-sensor. Another consideration is the battery life of smartwatches. Kheirkhahan et al. [42] indicated that the battery of smartwatches may be depleted due to additional sensors such as an accelerometer. In order to collect 24 h of continuous data, we did not want to lose data while the smartwatches were charging. The battery of smartwatches with other extra sensors such as an accelerometer do not last for more than one day or even 12 h; this would cause inconvenience to the subjects if the research staff needed to visit the subjects to replace smartwatches frequently for continuous monitoring. After considering all these factors, we chose the current model of smartwatch. It provides HR monitoring and a battery that lasts for 14 days. Our results showed that this model of smartwatch is an affordable complement to certified medical devices; it can be used by scientists in developing countries for PM_2.5_–health evaluation.

Some of smartwatches have an ECG HR monitor measuring HRV; these smartwatches are usually more expensive than those with a PPG sensor. However, to date, few studies have validated the HRV measurements obtained from the wrist-worn smartwatches with certified medical devices [43]. Therefore, more studies are needed to validate the HRV measurements directly obtained from wrist-worn smartwatches with certified medical devices under different activities. HRV can reflect the autonomic nervous system balance [44]. Some other types of smartwatches measure HRV with the aid of an extra chest strap, which may increase the discomfort of subjects, especially for 24 h continuous monitoring. Future research may adopt smartwatches with an ECG sensor or an additional chest strap, after validation, to assess the impacts of PM_2.5_ on HRV. For example, Apple Watch 6 is capable of providing HRV data but is about twice as expensive as Garmin Forerunner 35. For developing countries where resources are limited, low-cost smartwatches with a PPG sensor are still useful and affordable tools to assess the health impacts of PM_2.5_.

This study also has some limitations. First, only one brand of smartwatch was used to monitor the HR in this work. A previous work indicated that the HR monitoring accuracy may be different amongst the different brands of smartwatches [9]. Nevertheless, this work provides a successful example of the application of smartwatches with a PPG sensor to assess the PM_2.5_–HR relationship. Future research may evaluate the applicability of other brands of smartwatches in environmental health research. Next, we used the activity intensity measurements obtained from AS-LUNG-P, after converting to the RootiRx-comparable measurements via the correction equation, as an alternative to that for smartwatches. As mentioned above, the low correlation coefficients (*r* = 0.409) of the activity intensity measurements between AS-LUNG-P and RootiRx may have caused high variability. Nevertheless, our results still showed the same (positive) trend and a similar magnitude in the damage coefficients of PM_2.5_ on G-HR compared to those assessed via certified medical devices. Thus, the high variability did not affect our main findings. Finally, the stress may affect HR. We did not evaluate the psychosocial stress of subjects in this study. However, unless the impacts of psychological stress and PM_2.5_ exposure always occurred simultaneously, the psychological stress would not affect the estimated health damage coefficients of PM_2.5_ exposure in our models.

## 5. Conclusions

In this study, we successfully applied commercial smartwatches to evaluate the health impacts of PM_2.5_ and demonstrated the applicability of smartwatches in environmental health research. Compared to certified medical devices, the general population are more familiar with smartwatches, which minimally interfere with daily activities. Therefore, subjects can wear smartwatches for a longer monitoring period to increase the sample size available for health assessment. Although smartwatches may irregularly measure HR, the same (positive) trend and the magnitude of the health damage coefficients of PM_2.5_ as assessed via the smartwatch were similar to those assessed with a certified medical device. In addition, the findings demonstrated that during high-intensity activity periods, the impacts of PM_2.5_ might be stronger. These findings are interesting and important for the formulation of practical health-promotion tactics for people in exercising in polluted areas. However, due to higher variability and uncertainty associated with the measurements from smartwatches, we need more evidence to confirm the stronger impacts of PM_2.5_ on HR during exercising. With the advantages of low cost and low interference with daily activities, by applying smartwatches for HR monitoring, we can obtain larger HR sample sizes and thus further assess the health impacts of PM_2.5_ for different subpopulations in the future, which has been difficult with traditional instruments. Moreover, low-cost smartwatches together with PM_2.5_ LCS devices provide an opportunity to conduct PM_2.5_ health assessments in developing countries.

## Figures and Tables

**Figure 1 sensors-21-04585-f001:**
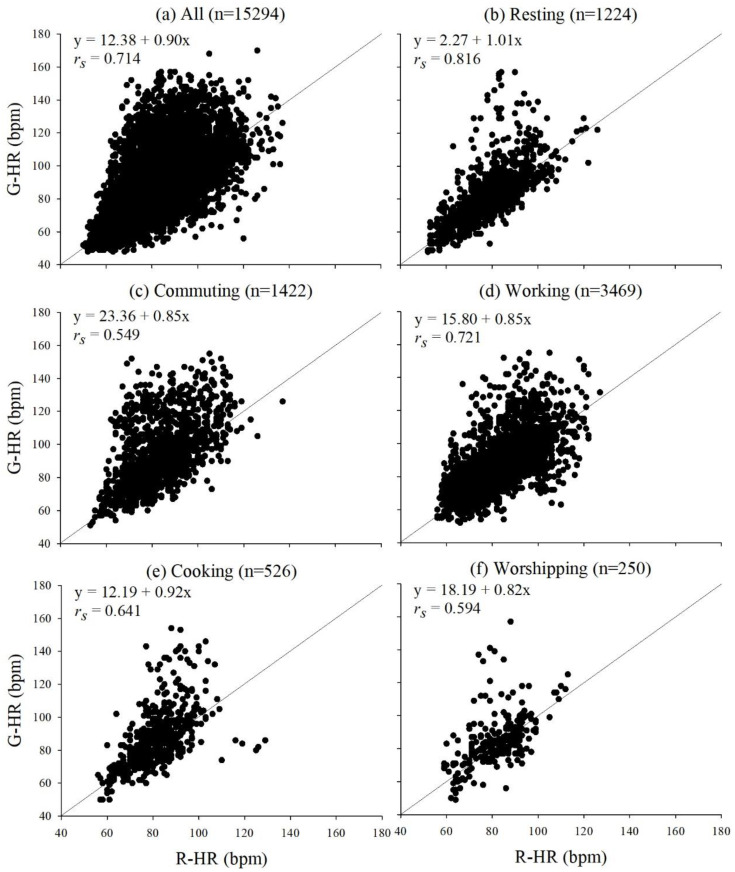
Correlations of heart rates between Garmin Forerunner 35 (G-HR) and RootiRx (R-HR) during various daily activities: (**a**) all activities, (**b**) resting, (**c**) commuting, (**d**) working, (**e**) cooking, (**f**) worshipping, (**g**) shopping, (**h**) exercising, (**i**) eating, (**j**) bath/shower, (**k**) sedentary activities, and (**l**) other activities.

**Figure 2 sensors-21-04585-f002:**
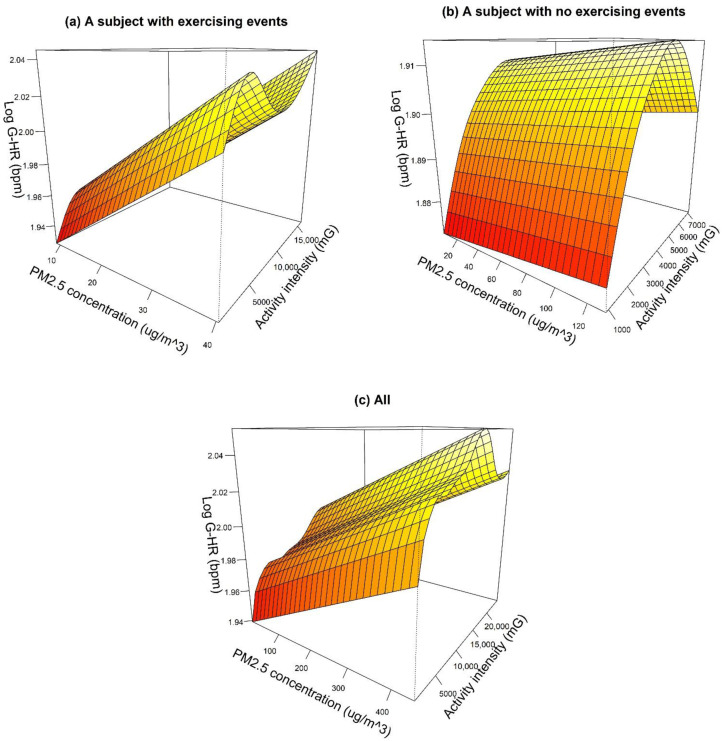
Relationships between PM_2.5_, activity intensity, and HR obtained from Garmin Forerunner 35 (G-HR) for (**a**) a subject with exercising events, (**b**) a subject without exercising events and (**c**) all subjects. The models were adjusted for temperature, activity, and time of day.

**Table 1 sensors-21-04585-t001:** (a) PM_2.5_ concentration, heart rate, and (b) activity intensity according to subjects’ characteristics.

**(a)**	**PM_2.5_ (μg/m^3^)**		**G-HR (bpm) ^a^**		**R-HR (bpm) ^b^**	
**Characteristics**	***n*^c^**	**Mean ± SD ^d^ (Midian)**	***n*^c^**	**Mean ± SD ^d^ (Midian)**	***n*^c^**	**Mean ± SD ^d^ (Midian)**
Age (years)						
40 to 64 years (27) ^e^	35,846	21.8 ± 12.8 * (20.3)	35,846	85.7 ± 16.2 * (83.0)	7731	83.0 ± 12.4 * (83)
65 to 75 years (22) ^e^	27,215	21.2 ± 9.8 (20.0)	27,215	84.2 ± 16.9 (82.0)	7563	80.9 ± 12.2 (80.0)
Gender						
Male (20) ^e^	24,844	21.4 ± 12.1 * (19.5)	24,844	85.0 ± 16.1 (82.0)	6170	83.0 ± 12.9 * (82.0)
Female (29) ^e^	38,217	21.6 ± 11.3 (20.6)	38,217	85.1 ± 16.8 (83.0)	9124	81.2 ± 11.9 (81.0)
Body mass index (BMI, kg/m^2^)						
<24 (19) ^e^	25,811	21.8 ± 10.0 * (20.5)	25,811	82.4 ± 15.8 * (80.0)	5811	78.4 ± 11.5 * (77.0)
≥24 (30) ^e^	37,250	21.3 ± 12.6 (20.0)	37,250	86.9 ± 16.7 (84.0)	9483	84.1 ± 12.3 (83.0)
**(b)**	**Activity intensity (AS-LUNG-P) ^f^ (mG ^g^)**			**Activity Intensity (RootiRx) ^h^ (mG ^g^)**		
**Characteristics**	***n*^c^**	**Mean ± SD ^d^ (Midian)**		***n*^c^**	**Mean ± SD ^d^ (Midian)**	
Age (years)						
40 to 64 years (27) ^e^	15,006 ^i^	2140 ± 1160 * (1920)		7731	1970 ± 450 * (1950)	
65 to 75 years (22) ^e^	10,963 ^i^	2090 ± 1070 (1900)		7563	1900 ± 430 (1890)	
Gender						
Male (20) ^e^	12,527 ^i^	1950 ± 1030 * (1950)		6170	1910 ± 440 * (1910)	
Female (29) ^e^	1344 ^i^	2280 ± 1180 (2090)		9124	1950 ± 440 (1920)	
Body mass index (BMI, kg/m^2^)						
<24 (19) ^e^	11,376 ^i^	2140 ± 1110 * (1960)		5811	1890 ± 410 * (1880)	
≥24 (30) ^e^	14,593 ^i^	2100 ± 1130 (1880)		9483	1960 ± 460 (1950)	

^a^ Heart rate data derived from Garmin Forerunner 35 through 7 day monitoring. ^b^ Heart rate data derived from RootiRx through 2 day monitoring. ^c^ Number of 5 min means of observations. ^d^ SD, standard deviation. ^e^ Numbers in parentheses are the number of subjects. ^f^ Activity intensity data derived from AS-LUNG-P through 7 day monitoring, which were converted to RootiRx-comparable data. ^g^ Milli-gravitational constant, 6.674 × 10^−14^ m^3^/kg s^2^. ^h^ Activity intensity data derived from RootiRx for 2 day monitoring. ^i^ Excluding activity intensity measurements below 1100 mG. * *p* < 0.001.

**Table 2 sensors-21-04585-t002:** Associations of 5 min means of PM_2.5_ concentration with R-HR ^a^ for 2 day monitoring (*n* = 15,294), and G-HR ^b^ for 7 day monitoring (*n* = 25,969).

	R-HR			G-HR		
	Coefficient ^c^	95% CI ^d^	*p*-Value	Coefficient ^c^	95% CI ^d^	*p*-Value
PM_2.5_	0.229	0.127, 0.332	<0.001	0.234	0.0801, 0.389	0.003
Age	−2.50	−8.23, 3.59	0.412	−1.83	−6.47, 3.04	0.454
BMI	5.78	−0.800, 12.8	0.086	6.50	1.33, 11.9	0.013
Gender	2.03	−4.23, 8.70	0.534	−1.31	−6.12, 3.74	0.604

^a^ Heart rate derived from RootiRx. ^b^ Heart rate derived from Garmin Forerunner 35 excluding activity intensity measurements <1100 mG. ^c^ Coefficients calculated as [10^(*β IQR*)^ −1] 100%, where β denotes the effect estimate. Coefficients expressed as percent change in HR associated with each interquartile range (IQR) increase in personal PM_2.5_ exposures, in models adjusted for subject, age, gender, body mass index (BMI), temperature, activity, and time of day. ^d^ CI, confidence interval.

**Table 3 sensors-21-04585-t003:** Associations of 5 min means of PM_2.5_ concentration with G-HR ^a^ during activities of various intensity through 7 day monitoring.

Activity	G-HR		
	Coefficient ^b^	95% CI ^d^	*p*-Value
Resting (*n* = 1296) ^c^	0.617	–0.117, 1.36	0.100
Low-intensity (*n* = 23909)	0.219	0.0606, 0.378	0.007
Moderate- to high-intensity (*n* = 764)	4.53	1.46, 7.70	0.004

^a^ Heart rate derived from Garmin Forerunner 35 excluding activity intensity measurements <1100 mG. ^b^ Coefficients calculated as [10^(*β IQR*)^ −1] 100%, where β denotes the effect estimate. Coefficients expressed as percent change in HR associated with each interquartile range (IQR) increase in personal PM_2.5_ exposure, in models adjusted for subject, age, gender, body mass index (BMI), temperature, activity, and time of day. ^c^ Data captured during sleeping were excluded. ^d^ CI, confidence interval.

## Data Availability

The data presented in this study are available on request from the corresponding author. The data are not publicly available due to privacy or ethical restrictions.

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
