# Peer review of "Demonstrating the Applicability of Smartwatches in PM2.5 Health Impact Assessment"

_sensors, 2021, doi:10.3390/s21134585_

Round 1

Reviewer 1 Report

It is a pleasure to review the manuscript entitled "Demonstrating applicability of smartwatches in health impact 2 assessment of PM2.5 by Tsou, Lung and Cheng.

The authors conducted experiments for assessing PM2.5 health impacts with wearable smart devices and certified medical devices and suggested that wearable smart devices could be alternatives. 

The manuscript was very well written. 

I only have a few suggestions and comments.

  1. It is better to use 'smartwatch' or 'smartwatches' throughout the manuscript (including the title).
  2. Please spell out 'CVD' for readers not familiar with it.
  3. It's better not to use 'AI' as an abbreviation to avoid confusion. Actually, instead of saying 'AI measurements', it is quite straightforward to say 'activity intensities' or 'activity intensity'. 
  4. In Table 1(b), the activity intensity measured by AS-LUNG-P shows a much higher standard deviation. Can the authors comment on that?   

Reviewer 2 Report

General comments

This paper is based on a one-week deployment of smart watches on 49 people to assess how heart rate (HR) as observed by these watches might be associated with PM levels reported by a portable PM monitoring package. During 2 of the days a reference level heart rate monitor was worn by the subjects to evaluate the HR measures from the watch. Activity levels were also assessed, primarily using diaries. A small increase in HR was found to be associated with PM.

There is a need for methods that allow the assessment of health impacts from microenvironmental pollutant levels encountered by people as they move through different areas and have differing levels of activity. However, a basic issue with this paper is that while it did include monitoring in a mobile population, the health metric (HR) is not well established in the paper as a marker of adverse impacts. Further, the watch selected may not be state-of-the-art and it did not include one important metric, that of acceleration which is related to activity assessments. Others might be found that collect more relevant data, perhaps including blood pressure, blood oxygen, and heart rate variability. Thus, while the study is a demonstration of the feasibility of use of smart watches in assessing interactions between pollution and HR, the findings are not compelling as a demonstration of a tool to study clearly adverse health outcomes from exposure. As a general observation, there seems to be a great deal of self-citation in many parts of the paper.

Specific comments

Line 19—what are “health damage coefficients”? This point is raised again, but not defined, on line 56 in the Introduction.

Line 44—mentions that watch monitors may be limited in ability to measure accurately during high level activities. This reviewers experience with one major offering of smart watches (Samsung Smartwatch 3) has been that they don’t fully function at all while motion is present. Please expand on this point and more clearly describe what activity levels are well assessed by the watch.

Line 87—Is the “AS-LUNG-P” a commercially available device? If so, please provide a more complete description/citation.

How was the Garmin selected from the various available smart watches? One limitation of this watch was that it did not include acceleration data which made it necessary to adjust data from a fraction of the monitoring period conducted. Are other watch devices available that do include this feature? A quick search on Google indicates that watches may be available with this feature.

Line 105—please expand the description of how sensor pack data were “converted to reference grade data”. How was the suitability of the lab calibrations confirmed vs. field conditions.

Line 121—please describe the data reduction and review process for data from the two day operations of the field ECG monitors. Was it completely automated or was qualified staff oversight included.  Current text simply states it was “uploaded and analysed using the empirical proprietary algorithms.”

Line 127—diary entries included “working”. What types of work were included in this group of subjects? In general the nature of the subjects is not adequately described in the methods section on line 77.

Line 148—it is unclear how robust the use for data from the 2 days of Rooti data was for deriving AI values for the week of AS-LUNG/ Garmin watch data.

Line 155—were the subjects allowed to place the AS device “on a nearby table” while at work? If so how might this impact the activity and exposure estimates? Overall, this brings up a point in studies that rely on self-reporting in diaries. How was compliance with protocol on AS device assured and how was compliance with diary needs assured?

What portion of HR data were captured? How was the capture rate distributed across various micro environments/activities? And in general, what was the rate of compliance with the entire protocol? What percent of subjects collected all data vs partial data?

Figure 1—2nd best fit as seen in graph of “other”, even though there are few observations. Please explain this. Further, please discuss the implications of having very high agreement  for “all” vs. any specific activities.

Further for figure 1, it seems that x and y axes are scaled differently. It might be useful to alter the plots in two ways. The figure would be more useful if scaled the same, provided the slopes or full equations for the fits observed for each panel and included 1:1 lines.

What is the difference between “resting” and “sedentary” as shown in separate panels of the figure?

Section 3.3—associations between HR and PM are presented. However, it would be helpful in preceding text to describe what is known regarding the health impacts of various levels of HR or heart rate change. Is HR an accepted metric of health impact vs. other related factors such as HR variability? In this case the changes of HR are quite small. The example given was for a 0.282% change in HR per 10ug/m3 PM 2.5. Line 70 of the paper mentions “estimated damage coefficients”. What is the evidence that small HR changes are associated with established health “damage”? Again on line 292 there is inference of adverse effects by the use of the undefined “damage coefficient”.

Line 302—there is consideration of lower agreement between HR methods during exercise. However, the paper should contain explicit information regarding the factors involved. Specially, what was the amount of successfully captured HR data from the watch during exercise?

Overall, findings of smartwatch for HR increases vs PM levels seem small, but pretty well demonstrated. Thus, these findings are novel. The implications of these findings are not established. Other measures could/should have been considered that do have accepted health implications. These include HRV. This issue is included first on line 389. This reviewer has a wrist mounted HRV sensor and it was not costly. This point is relevant when the utility of the method is covered by discussion text as is seen in the two following specific lines of the text:

line 334—statements related to the general application of the approach are made that are over reaching--”This work demonstrated that low-cost smart watches are useful alternatives to medical devices in environmental health studies…”. Again, on line 376-- “This again demonstrates the feasibility of applying smart watches in environmental health research.” and

Line 336—the utility of these low cost methods in developing countries may be limited by difficulties with calibration of PM monitors as with variable humidity impacts.

A final and relevant point on the consideration of devices for future use is that as the authors point out there has been considerable expansion of offerings by smart watch makers. Would other watches do a better job?

Line 442 in conclusions--”this study demonstrated that, during high-intensity activity periods, the impact of PM2.5 is one order of magnitude greater.” This statement needs some clear cautions based on the study and a thoughtful discussion of what health risks of a small HR might be.  And since the paper does mention that watches may or may not function well during high intensity activity it is not clear that the point is fully supported. Please rephrase these points with details and cautions.

Reviewer 3 Report

This topic of this study is interesting. However, I have some concerns on the developed equation and low correlation coefficients on PM2.5 concentration and R-HR and G-HR, as shown in the manuscript. Further justifications are needed for the physical implications of these equations and coefficients. In addition, solid evidences are needed for the influence of short-term exposure of PM2.5 to the HR. Additional contributions from daily activities (e.g., leisure activities) to HR should also be considered and discussed in the manuscript. 

Round 2

Reviewer 3 Report

The manuscript is significantly improved.

Author Response

Point 1: The manuscript is significantly improved.

Response 1: Thank you very much for your support. We are very grateful for your insightful comments and helpful suggestions. Those comments are all valuable and very helpful for revising and improving our paper.